# Training-Free Pseudo-Fusion Strategies for Composed Image Retrieval via Diffusion and Multimodal Large Language Models

## Abstract

Composed Image Retrieval (CIR) is an emerging paradigm in content-based image retrieval that enables users to formulate complex visual queries by combining a reference image with an auxiliary modality, usually text-based. This approach supports fine-grained search where the target image shares structural elements with the user query but is modified according to the provided auxiliary text. Conventional CIR methods rely on multimodal fusion to combine visual and textual features into a joint query embedding. In this work, we propose PEFUSE (for pseudo-fusion), a training-free framework that leverages pretrained models to bridge modalities via generative conversion. We introduce two novel strategies: uni-directional and bi-directional conversion, both implemented using diffusion models and multimodal large language models. These methods reformulate CIR as either intra-modal or cross-modal retrieval, bypassing the need for dedicated training. Extensive experiments on standard benchmarks show that our approach achieves competitive or superior performance compared to state-of-the-art methods, highlighting the efficacy and flexibility of our pseudo-fusion paradigm for composed retrieval. Our code is available at: `https://github.com/TBA`.

## 1 Introduction

Traditional Content-Based Image Retrieval systems allow users to submit image-based queries, bridging the so-called *semantic gap* (Smeulders et al., 2000). This constraint hinders their ability to accommodate nuanced search intents that are inherently multimodal. Composed Image Retrieval (CIR) addresses this limitation by enabling users to formulate a query using a reference image coupled with an auxiliary modality to specify desired modifications, often a natural language description. This approach facilitates fine-grained retrieval, such as finding "this chair but in blue" or "the same scene at sunset," which is particularly valuable in domains like e-commerce (Baldrati et al., 2022), digital asset management (Net & Gomez, 2025), and creative design (Song et al., 2025).

CIR introduces distinct technical challenges. An effective system must not only comprehend the individual modalities but also model their compositional semantics, capturing how the auxiliary input alters the meaning of the reference image. A prevalent solution involves multimodal feature fusion, in which visual and auxiliary representations are integrated into a unified embedding prior to retrieval. Although recent advances in deep convolutional and transformer-based architectures (Vaswani et al., 2017; Dosovitskiy et al., 2021) have improved cross-modal alignment and compositional reasoning, the majority of existing methods rely heavily on dedicated training on large-scale, annotated CIR datasets. To relieve the restrictions, researcher either synthesize triplet datasets (Li et al., 2025; Wang et al., 2025; Xing et al., 2025) or rely on existing image-text pairs (Jiang et al., 2024) to train models. This dependency limits their scalability and adaptability to significant domain shifts, wherein zero-shot CIR is regarded as an effective solution.

In this work, we investigate *training-free pseudo-fusion* strategies for zero-shot CIR that circumvent the need for additional task-specific fusion. We propose to pseudo-fuse the multimodal query through *uni-directional* and *bi-directional conversion* techniques, leveraging recent advancements in Diffusion Models (Ho et al., 2020; Song et al., 2021; Rombach et al., 2022) and Multimodal Large Language Models (MLLMs) (Liu et al., 2023; Grattafiori et al., 2024; Yang et al., 2025). The uni-directional approach reformulates the multimodal query CIR task into a standard uni-modal

query problem by converting the reference image and auxiliary text into a synthesized image or a detailed textual description. The bi-directional approach extends this by additionally converting the candidate images in the gallery into texts, enabling a text-based matching paradigm. These strategies facilitate flexible adaptation of existing, off-the-shelf retrieval systems without requiring architectural modifications, fine-tuning, or any training.

Extensive experiments on standard CIR benchmarks demonstrate that our proposed training-free methods achieve competitive or superior performance compared to the state-of-the-art (SOTA) trained models and other training-free methods. Our findings underscore the significant potential of training-free approaches in compositional retrieval when deploying efficient CIR systems in resource-constrained or rapidly evolving domains. To the best of our knowledge, this is the first work to systematically explore and benchmark modality conversion strategies utilizing both diffusion models and MLLMs for CIR. In summary, our contributions are as follows:

- Novel training-free pseudo-fusion strategies for zero-shot CIR that seamlessly convert multi-modal queries into a single modality, enabling compatibility with existing retrieval systems.
- A systematic study and comprehensive benchmarking of both uni-directional and bi-directional modality conversion paradigms for CIR using diffusion models and MLLMs.
- A quantitative analysis to elucidate the relationship between CIR performance and key hyperparameters of both MLLMs and diffusion models.
- A component-based computational analysis when resorting MLLMs and diffusion models when reframing CIR task to single modality retrieval tasks.
- Empirical evidence that reformulating CIR to text-to-image retrieval is more effective than other tasks, and that our method achieves on-par or better performance than SOTA models.

## 2 RELATED WORK

### 2.1 TRAINING-DEMANDING COMPOSED IMAGE RETRIEVAL METHODS

Early CIR approaches like TIRG (Vo et al., 2019) relied on joint embedding spaces trained with contrastive objectives (van den Oord et al., 2018; Chen et al., 2020; He et al., 2020), where the fused image–text representation was directly compared against candidate image embeddings. Subsequent transformer-based methods (Jia et al., 2021; Li et al., 2022; 2023), pretrained on large-scale vision–language datasets, achieved stronger cross-modal alignment. Building on this foundation, Combiner (Baldrati et al., 2022) leverages CLIP (Radford et al., 2021) to compute integrated features from reference images and accompanying textual descriptions.

A notable line of work builds upon the idea of representing images as pseudo-word tokens within a text sequence. Inspired by Textual Inversion (Gal et al., 2023), methods such as SEARLE (Baldrati et al., 2023b), Pic2Word (Saito et al., 2023), and LinCIR (Gu et al., 2024) map reference images into token embeddings that can be processed by language models, achieving SOTA performance through joint training. Other approaches like CLIP4CIR (Baldrati et al., 2024) introduce learnable fusion operators to better capture compositional semantics.

More recently, the generative capabilities of diffusion models (Ho et al., 2020; Song et al., 2021; Rombach et al., 2022) have also been explored for CIR. For example, CIG (Wang et al., 2025) uses a pretrained textual inversion network to convert a reference image into tokens and employs a latent diffusion model to generate a visual representation of the target image, which is then fused with the query. Despite these advances, the use of LLMs or MLLMs remains more prevalent. For example, DQU-CIR (Wen et al., 2024) fuses unified textual and visual information extracted via LLMs or captioning models. Notably, HyCIR (Jiang et al., 2024) enhances model training by incorporating contrastive learning on synthetic triplets, demonstrating the efficacy of synthetic supervision. Further advancing this approach, Feng et al. (2024) scales both negative and positive samples for contrastive learning using a MLLM. Furthermore, MRA-CIR (Tu et al., 2025) circumvents error-prone intermediate text generation by using a Multimodal Reasoning Agent to directly construct high-quality triplets from unlabeled images. Similarly, CoLLM (Huynh et al., 2025) mitigates data scarcity by synthesizing training triplets from image-caption pairs using LLMs, enabling deeper multimodal fusion.

A common characteristic of all the above methods is their reliance on ad-hoc training, either fine-tuning on synthetic triplets or training from scratch on existing image-text datasets. In contrast, our approach requires no training. Instead of synthesizing training data or inverting images to tokens, we convert multimodal queries into real text tokens (via MLLMs) or real images (via diffusion models), making them directly compatible with existing retrieval models. This allows implicit fusion of both modalities using purely pretrained models, in a fully training-free manner.

## 2.2 TRAINING-FREE COMPOSED IMAGE RETRIEVAL METHODS

Although learnable fusion methods achieve strong performance, their requirement for task-specific training limits flexibility and generalizability to new domains or modalities. To overcome these limitations, several training-free CIR methods have been proposed. CIReVL (Karthik et al., 2024) uses an LLM to refine captions (generated by a vision-language model from a reference image) by incorporating the text modification. The resulting caption is then used for text-to-image retrieval. Similarly, WeiMoCIR (Wu et al., 2025) employs an MLLM to caption candidate target images, effectively reducing CIR to a matching problem. Our method extends this idea by using the MLLM to process both the reference image and the modification text, effectively using the MLLM as a fusion module. Another relevant framework, ImageScope (Luo et al., 2025), unifies various language-guided image retrieval tasks into a text-to-image retrieval setup using descriptions generated by a MLLM. However, it relies on multiple models applied in stages, leading to cumulative error propagation and increased inference time. In contrast, our approach uses only a single MLLM or diffusion model, resulting in a simpler yet effective pipeline.

Unlike these prior works, which often involve multi-stage text generation or ensemble multiple models, we demonstrate that a single MLLM can effectively capture interactions between the reference image and modification text to produce informative textual descriptions of the desired target. Furthermore, we systematically explore and benchmark alternative formulations of CIR, including conversion to one-query intra-modal or cross-modal retrieval tasks. Specifically, our work introduces training-free pseudo-fusion strategies that reformulate CIR as either text-based or image-based retrieval. By leveraging pretrained diffusion models and MLLMs without any additional training, our approach offers a flexible, modular, and plug-and-play solution for composed image retrieval.

## 3 METHODOLOGY

As depicted in Figure 1, our method employs a dual-strategy pipeline. The uni-directional conversion facilitates retrieval by projecting the query into a target modality (purple dashed box); either by using an MLLM to generate descriptive texts from images and modifications or by using a diffusion model to generate images from reference images and MLLM-generated texts. Specifically, we use MLLMs to convert reference images plus corresponding modifications to texts or use diffusion models to generate alternative target images based on such composed queries. The bi-directional conversion extends this by subsequently using the MLLM to also project target images into the textual modality, enabling a text-based retrieval process (green dashed box). Namely, we additionally generate texts based on target images, and then match with texts or images from uni-directional conversion.

Let $\mathcal{I}$ and $\mathcal{T}$ be the image and text spaces, respectively. Assume we have a retrieval model $\Psi(\cdot)$ that takes both image modality $I \in \mathcal{I}$ and text modality $T \in \mathcal{T}$ as inputs, and outputs a similarity score $s = sim(\Psi(I), \Psi(T)) \in \mathbb{R}$ based on extracted embeddings $\Psi(I) \in \mathbb{R}^m$ and $\Psi(T) \in \mathbb{R}^m$, a MLLM $f(\cdot)$ which can generated textual descriptions $T_f$ based on arbitrary combination of images $I$ and texts $T$ given the proper dataset-specific prompt $p$, and a diffusion model $g(\cdot)$ that generates image $I_g$ based on both images $I$ and texts $T$. For CIR, a reference image $I_{ref}^i \in \mathcal{I}$ with corresponding text modification $T_i \in \mathcal{T}$ are paired as a paired query $(I_{ref}^i, T^i)$, to find the most relevant candidate images in dataset $\mathcal{D} = \{I_{tar}^1, I_{tar}^2, \ldots, I_{tar}^n\}$ based on (cosine) similarity scores.

For uni-directional conversion, we use the MLLM $f$ to fuse $(I_{ref}^i, T^i)$: $T_f^i = f(I_{ref}^i, T^i, p) \in \mathcal{T}$ while diffusion model $g$ is used to synthesize images based on MLLM-generated descriptions $T_f^i$: $I_g^i = g(I_{ref}^i, T_f^i) \in \mathcal{I}$. To use pretrained retrieval models, we feed the generated texts $T_f^i$ to retrieval model $\Psi$ to compute cosine similarity with respect to all candidate images. $s_{ij} =$

$\cos(\Psi(T_f^i), \Psi(I_{tar}^j))$, Through this method, we reformulate CIR to two types of image retrieval tasks: text-to-image and image-to-image.

For bi-directional conversion, we additionally convert target images $I_{tar}$ to texts via MLLM given another dataset-specific prompt $q$: $T_f^j = f(I_{tar}^j, q)$ and compute the cosine similarity based on previously generated modalities, either by $s_{ij} = \cos(\Psi(T_f^i), \Psi(T_f^j))$ or $s_{ij} = \cos(\Psi(I_g^i), \Psi(T_f^j))$ for $j \in \{1, \ldots, n\}$, which reformulates the CIR tasks to text-to-text retrieval and image-to-text retrieval tasks, respectively. Based on the ranked scores, we return the top-$K$ candidate image IDs for performance evaluation.

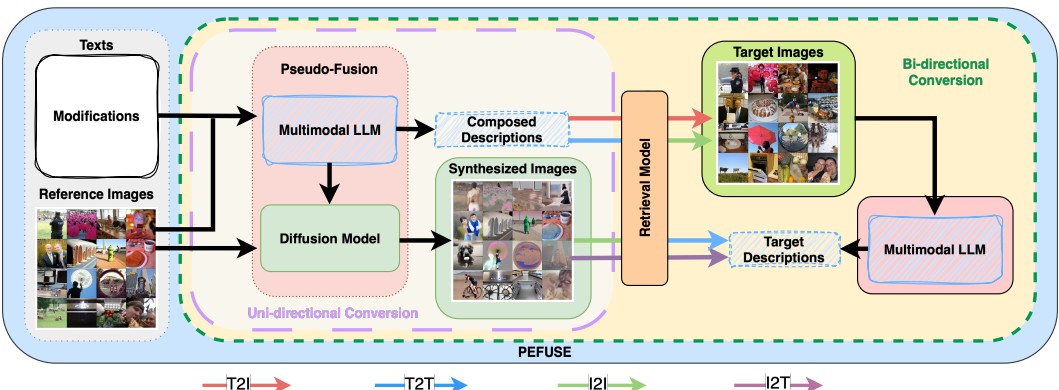

Figure 1: Our proposed training-free pseudo-fusion methods for Composed Image Retrieval. Dashed green box indicates bi-directional conversion, while the dashed purple box is uni-directional conversion. T2I: text-to-image; T2T: text-to-text; I2T: image-to-text; I2I: image-to-image.

Our method is pseudo-fusion as it relies on generative models to synthesize new data (images or text) from pairs of elements within a triplet, thereby achieving an implicit fusion of modalities. This approach is distinct from typical early fusion paradigms, which explicitly combine modalities into intermediate embeddings. In contrast, our method directly generates coherent and interpretable data in a target modality, preserving latent semantics throughout the process.

## 4 EXPERIMENTS

We first state the experiment setups such as datasets and models used, and then present results after evaluating these models for different conversion methods.

### 4.1 DATA AND MODELS

We employ four CIR benchmark datasets: Fashion-IQ (Wu et al., 2021), CIRR (Liu et al., 2021), CIRCO (Baldrati et al., 2023a), and GeneCIS (Vaze et al., 2023). Fashion-IQ is designed for interactive fashion image retrieval using natural language feedback, incorporating human-written relative captions and derived visual attributes. CIRR extends the scope to open-domain images with human-annotated modifying text, though it is known to contain a significant number of false negatives (Baldrati et al., 2023a). To mitigate this issue, CIRCO provides multiple ground-truth images per query, with all images sourced from the MS-COCO (Lin et al., 2014) dataset. In addition, GeneCIS measures models' ability to adapt to a range of similarity conditions in terms of attributes and objects. In line with standard evaluation protocols, we report recall@$K$ for Fashion-IQ, CIRR, and GeneCIS, and mean average precision (mAP@$K$) for CIRCO, reflecting their respective annotation structures.

For image synthesis based on textual and visual inputs, we utilize the pretrained `SDXL-InstructPix2Pix` model from the `diffusers` library, an instruction-tuned variant of InstructPix2Pix (Brooks et al., 2023), selected for its strong generative performance. Text generation is handled by `Qwen2.5-VL-7B-Instruct`, an advanced instruction-tuned MLLM based on Qwen (Yang et al., 2025). We later use other MLLMs and diffusion models to analyze computational overhead incurred by our pipeline in section 5. To evaluate the effectiveness of our approach, we

employ several retrieval models as feature extractors, starting with models sharing the same backbone architecture: CLIP (ViT-B/32) (Radford et al., 2021), OpenCLIP (ViT-B/32) (Cherti et al., 2023), and SigLIP2 (Base-Patch16) (Tschannen et al., 2025). Both CLIP and OpenCLIP adopt the softmax function in their contrastive loss formulations, with OpenCLIP additionally benefiting from training on substantially larger datasets. In contrast, SigLIP2 incorporates several enhancements to improve semantic understanding over SigLIP (Zhai et al., 2023) that was trained with sigmoid-based contrastive loss function. We later explore larger model variants to investigate scaling behavior.

All retrieval models operate on input images resized to $224 \times 224$ pixels, normalized to the $[0, 1]$ range using model-specific normalization parameters. The diffusion model requires $768 \times 768$ pixel inputs and produces outputs at the same resolution. For consistency, all images across datasets are resized to $768 \times 768$ and normalized to $[0, 1]$ prior to diffusion processing. During image generation with the diffusion model, we use a guidance scale of 7.5, image guidance scale of 3.0, and 30 denoising steps. For text generation with the MLLM, we set the temperature to 0.1, top-$P$ to 0.9, and top-$K$ to 50. A sensitivity study in subsection 4.5 examines the impact of varying these hyperparameters. Our implementation uses `PyTorch` on a single NVIDIA A100 with 40GB memory.

## 4.2 UNI-DIRECTIONAL CONVERSION

As introduced in section 3, uni-directional conversion can be implemented using either MLLMs or diffusion models, and both serve as pseudo-fusion methods.

Table 1 presents the text-to-image and image-to-image retrieval results on the Fashion-IQ dataset. Among the compared zero-shot methods, most require training new models, with CIReVL being the notable exception. When reformulating CIR as a text-to-image retrieval task, both CLIP and OpenCLIP—utilizing the ViT-B/32 backbone—outperform CIReVL. Notably, employing OpenCLIP as the retriever surpasses the performance of most zero-shot methods that necessitate model training, with results comparable to, though slightly lower than, LinCIR, which uses larger ViT-L/14 backbone.

In general, text-to-image retrieval demonstrates superior performance compared to image-to-image retrieval; the sole exception is observed with the SigLIP2 model. Across all retrieval models evaluated, we note a significant inconsistency in performance: SigLIP2 yields the weakest results for text-to-image retrieval, yet achieves the strongest performance for image-to-image retrieval. This disparity underscores the substantial variation in semantic understanding capabilities among retrieval models on varying tasks, highlighting their critical and impactful role in the effectiveness of CIR systems.

Table 1: Performance (%) comparison on **Fashion-IQ** validation split using different retrieval models via PEFUSE to convert reference images and modifications to composed text or synthesized images. All the retrieval models use ViT-B/32 backbone. Best results in boldface while the second best underscored. $^*$: reproduced results; $^\dagger$: results from original papers.

| Method | Retrieval Model | Shirt | | Dress | | Toptee | | Average | |
|---|---|---|---|---|---|---|---|---|---|
| | | R@10 | R@50 | R@10 | R@50 | R@10 | R@50 | R@10 | R@50 |
| Pic2Word$^\dagger$ | CLIP (ViT-L/14) | 26.20 | 43.60 | 20.00 | 40.20 | 27.90 | 47.40 | 24.70 | 43.70 |
| SEARLE-OTI$^*$ | CLIP (ViT-B/32) | 24.43 | 41.39 | 19.85 | 40.72 | 24.85 | 45.47 | 23.05 | 42.53 |
| SEARLE$^*$ | CLIP (ViT-B/32) | 24.85 | 41.60 | 19.37 | 39.21 | 25.12 | 46.22 | 23.11 | 42.34 |
| CIReVL$^*$ | CLIP (ViT-B/32) | 18.40 | 30.82 | 14.25 | 30.45 | 18.00 | 34.33 | 16.88 | 31.87 |
| CoLLM$^\dagger$ | CLIP (ViT-B/32) | 24.90 | 45.10 | 22.90 | 43.80 | 26.40 | 46.80 | 24.80 | 45.20 |
| LinCIR$^*$ | CLIP (ViT-L/14) | **29.69** | **48.14** | 22.32 | **45.13** | 30.85 | 52.01 | 27.62 | **48.43** |
| SEARLE+CIG-XL turbo$^\dagger$ | CLIP (ViT-B/32) | 24.73 | 41.46 | 18.94 | 39.66 | 25.50 | 46.66 | 23.06 | 42.59 |
| HyCIR$^\dagger$ | CLIP (ViT-L/14) | 27.62 | 44.94 | 19.98 | 40.80 | 28.14 | 47.67 | 25.25 | 44.47 |
| | CLIP | 20.62 | 37.11 | 13.99 | 32.54 | 19.93 | 39.58 | 18.18 | 36.41 |
| PEFUSE (T→I) | OpenCLIP | 28.30 | 46.19 | **24.05** | 44.11 | **32.46** | **53.94** | **28.27** | 48.08 |
| | SigLIP2 | 6.49 | 13.61 | 7.32 | 17.05 | 7.23 | 16.93 | 7.01 | 15.86 |
| | CLIP | 8.97 | 17.01 | 4.95 | 13.82 | 7.71 | 16.76 | 7.21 | 15.87 |
| PEFUSE (I→I) | OpenCLIP | 14.28 | 24.48 | 10.33 | 22.43 | 12.69 | 24.37 | 12.43 | 23.76 |
| | SigLIP2 | 15.00 | 26.39 | 9.25 | 20.93 | 13.77 | 25.01 | 12.67 | 24.11 |

The retrieval performance on the CIRR and CIRCO datasets is further detailed in Table 2 and Table 3, respectively. On the CIRR dataset, for the text-to-image retrieval task, the CLIP model slightly

underperforms compared to other methods wile being notably better on CIRR subsets. In contrast, SigLIP and OpenCLIP achieve significantly stronger performance, on both the CIRR and its subsets. For the image-to-image task on CIRR, all retrieval models fall behind the established baselines, underscoring the superiority of reformulating CIR as a text-to-image rather than an image-to-image retrieval task.

A similar phenomenon is observed on the CIRCO dataset, where text-to-image retrieval outperforms baseline methods, while the baselines surpass image-to-image conversion. Specifically, text-to-image retrieval using CLIP performs competitively, exceeding training-free methods such as CIReVL, though it remains behind LinCIR and HyCIR, both of which employ a larger ViT-L/14 backbone and demand training. Notably, within the same task, using OpenCLIP and SigLIP with a ViT-B/32 backbone surpasses all baseline methods by a considerable margin. This indicates that employing a more powerful retrieval model can substantially enhance system performance. Conversely, experiments on image-to-image retrieval for CIRCO demonstrate inferior results, highlighting a need for improvement in diffusion-based conversion methods. We show results on GeneCIS in Table 7 in Appendix C, which further corroborates our analysis.

The experimental results across all datasets demonstrate that our method is effective for the zero-shot CIR task, despite its simplicity and training-free nature. Although CLIP has been widely adopted in previous studies, our results indicate that it is a suboptimal choice for CIR systems compared to OpenCLIP. We further note that methods which separately generate captions via an image captioner and then combine them with modification text using a LLM (e.g., CIReVL) can be effective for simple images, such as fashion items. However, in complex scenarios like those in CIRCO, which involve a large pool of candidate images (123K), such pipelines often fail to adequately capture the intricate interactions between reference images and textual modifications. This leads to inferior retrieval performance compared to ours. Consequently, employing a MLLM proves to be both sufficient and less error-prone, outperforming lengthy, chained pipelines for complex image retrieval task.

Table 2: Performance (%) comparison on **CIRR** test split using different retrieval models via PEFUSE to convert reference images and modifications to composed texts or synthesized images. All the retrieval models use ViT-B/32 backbone. Best results in boldface while the second best underscored. †: results from original papers; ∗: reproduced results; —: results not available.

| Method | Retrieval Model | Recall | | | | | Recall$_{subset}$ | | |
|---|---|---|---|---|---|---|---|---|---|
| | | @1 | @2 | @5 | @10 | @50 | @1 | @2 | @3 |
| Pic2Word† | CLIP (ViT-L/14) | 23.90 | — | 51.70 | 65.30 | 87.80 | 53.76 | 74.46 | 87.08 |
| SEARLE-OTI∗ | CLIP (ViT-B/32) | 23.18 | 34.72 | 52.31 | 66.00 | 89.21 | 52.02 | 74.43 | 86.75 |
| SEARLE∗ | CLIP (ViT-B/32) | 23.33 | 34.89 | 52.89 | 66.99 | 89.81 | 53.90 | 76.19 | 87.76 |
| CIReVL∗ | CLIP (ViT-B/32) | 21.40 | 31.86 | 47.74 | 60.72 | 84.99 | 56.27 | 77.08 | 88.63 |
| CoLLM† | CLIP (ViT-B/32) | 28.60 | — | — | 71.80 | 92.70 | — | — | — |
| LinCIR∗ | CLIP (ViT-L/14) | 25.04 | 36.22 | 53.78 | 67.18 | 88.75 | 56.53 | 76.82 | 88.70 |
| SEARLE+CIG-XL turbo† | CLIP (ViT-B/32) | 25.54 | — | 55.01 | 68.24 | 90.72 | 57.52 | 78.36 | 89.35 |
| HyCIR† | CLIP (ViT-L/14) | 25.08 | — | 53.49 | 67.03 | 89.85 | 53.83 | 75.06 | 87.18 |
| PEFUSE (T→I) | CLIP | 22.58 | 32.96 | 49.81 | 63.59 | 87.47 | 63.28 | 82.27 | 91.76 |
| | OpenCLIP | **34.00** | **47.49** | **65.57** | **77.47** | **93.28** | **71.59** | **88.39** | **95.04** |
| | SigLIP2 | 30.65 | 43.13 | 60.60 | 72.43 | 91.35 | 70.00 | 86.48 | 93.64 |
| PEFUSE (I→I) | CLIP | 2.77 | 9.78 | 23.06 | 35.42 | 64.58 | 29.90 | 52.19 | 70.96 |
| | OpenCLIP | 3.28 | 11.64 | 27.76 | 41.45 | 71.37 | 31.01 | 53.83 | 72.12 |
| | SigLIP2 | 3.78 | 12.72 | 28.39 | 42.12 | 70.07 | 32.43 | 54.15 | 71.35 |

## 4.3 BI-DIRECTIONAL CONVERSION

Besides leveraging a MLLM to fuse the information from reference images and their corresponding modification texts into unified textual descriptions, now the same model is additionally employed to generate descriptive captions for target images. This methodology effectively reformulates the CIR task into a text-to-text retrieval problem. Together with generated images via diffusion models, CIR task is reframed as image-to-text retrieval task. The performance of text-retrieval-based conversion is presented in Appendix C and Appendix E.

We observe that reformulating CIR as a text-to-text retrieval task generally yields stronger performance compared to image-to-text retrieval. We hypothesize that this is due to artifacts in gen-

Table 3: Performance (%) comparison on **CIRCO** test split using different retrieval models via PEFUSE to convert reference images and modification texts to composed texts or to synthesized images. All the retrieval models use ViT-B/32 backbone. Best results in boldface while the second best underscored. ‡: results from CIReVL; ∗: reproduced results; —: results not available.

| Method | Retrieval Model | mAP@5 | mAP@10 | mAP@25 | mAP@50 |
|---|---|---|---|---|---|
| Pic2Word‡ | CLIP (ViT-L/14) | 8.72 | 9.51 | 10.64 | 11.29 |
| SEARLE-OTI∗ | CLIP (ViT-B/32) | 7.29 | 7.99 | 9.21 | 9.85 |
| SEARLE∗ | CLIP (ViT-B/32) | 9.38 | 9.95 | 11.13 | 11.85 |
| CIReVL∗ | CLIP (ViT-B/32) | 10.36 | 10.70 | 11.88 | 12.47 |
| CoLLM† | CLIP (ViT-B/32) | 12.90 | 13.20 | — | 15.00 |
| LinCIR∗ | CLIP (ViT-L/14) | 12.33 | 13.13 | 14.56 | 15.46 |
| SEARLE+CIG-XL turbo† | CLIP (ViT-B/32) | 10.45 | 11.02 | 12.34 | 13.00 |
| HyCIR† | CLIP (ViT-L/14) | 14.12 | 15.02 | 16.72 | 17.56 |
| PEFUSE (T→I) | CLIP | 11.12 | 11.47 | 12.86 | 13.61 |
| | OpenCLIP | 16.89 | 17.56 | 19.14 | 20.16 |
| | SigLIP2 | **18.53** | **19.68** | **21.58** | **22.62** |
| PEFUSE (I→I) | CLIP | 2.39 | 2.57 | 3.08 | 3.37 |
| | OpenCLIP | 3.03 | 3.49 | 4.10 | 4.44 |
| | SigLIP2 | 4.06 | 4.63 | 5.47 | 5.94 |

erated images, which introduce a larger semantic gap between modalities for retrieval models, whereas texts generated by MLLMs retain more semantically meaningful information. Furthermore, as shown in subsection 4.2, both text-to-text and image-to-text retrieval underperform relative to text-to-image retrieval. However, image-to-text retrieval generally surpasses image-to-image performance. These findings further instantiate that reformulating CIR as a text-to-image retrieval task is generally more effective than other conversion strategies under the same settings.

## 4.4  SCALING LAW

Having evaluated our method's performance using retrieval models with a ViT-B/32 backbone in subsection 4.2, a subsequent question arises regarding the potential benefits of larger backbone architectures. To investigate this, we assess the performance of OpenCLIP—selected for its superior overall performance among the three retrieval models—using ViT backbones of varying sizes across all datasets. The overall results are presented in Table 4, with detailed category-specific results for Fashion-IQ and CIRR subsets provided in Table 8 in Appendix D. As shown in Table 4, we observe a general trend of improving performance for the text-to-image retrieval task as the backbone size increases, although performance fluctuates across specific model sizes. Notably, on Fashion-IQ, Recall@10 decreases and Recall@50 saturates when using the ViT-g/14 backbone. Performance on CIRR also improves consistently with model scale, with the exception of a slight decrease in Recall@1 and Recall@10 for ViT-g/14. A similar performance drop with ViT-g/14 is observed on the CIRCO dataset. Furthermore, this trend of scaling benefits extends beyond text-to-image retrieval; larger models consistently achieve superior performance on image-to-image, text-to-text, and image-to-text retrieval tasks as well when using our proposed methods to reformulate CIR task. Similar trend holds for GeneCIS. Category results of GeneCIS can be found in Table 9 in Appendix D.

Table 4: Scaling law when using different backbones for OpenCLIP for text-to-image retrieval on each benchmark.

| Backbone | Fashion-IQ | | CIRR | | | | | CIRCO | | | | GeneCIS |
|---|---|---|---|---|---|---|---|---|---|---|---|---|
| | R@10 | R@50 | R@1 | R@2 | R@5 | R@10 | R@50 | mAP@5 | mAP@10 | mAP@25 | mAP@50 | R@1 |
| ViT-L/14 | 29.49 | 48.37 | 36.17 | 50.07 | 67.13 | 78.72 | 94.17 | 21.69 | 22.99 | 25.07 | 26.14 | 16.85 |
| ViT-H/14 | 30.40 | 50.09 | 38.55 | 52.02 | 69.49 | 80.29 | 94.29 | 23.78 | 24.78 | 27.10 | 28.24 | 17.50 |
| ViT-g/14 | 30.15 | 50.12 | 38.41 | 52.15 | 70.15 | 80.27 | 94.58 | 22.62 | 23.93 | 26.39 | 27.42 | 17.39 |
| ViT-bigG/14 | 30.44 | 49.49 | 40.41 | 54.63 | 71.13 | 81.06 | 94.89 | 25.03 | 26.63 | 29.17 | 30.28 | 17.99 |

## 4.5 SENSITIVITY ANALYSIS

We study how the model performance can be impacted by varying values for hyperparameters of the MLLM and the diffusion model we used. To the best of our knowledge, this is the first time that relationship between CIR performance and these hyperparameters is studied.

### 4.5.1 MULTIMODAL LARGE LANGUAGE MODELS

For MLLM, temperature controls the determinism of the LLM when generating tokens, Top-$P$ is the probability that model selects tokens up to probability $P$, and Top-$K$ sampling limits the model to consider only the k most likely tokens at each step. For consistency with previous experiments, we use 0.1 for temperature, 0.9 for top-$P$, 50 for top-$K$ as base combination and only alter one parameter while the other are fixed. For example, when we investigate how temperature would impact the retrieval performance, we fix top-$P$ to 0.9 and top-$K$ to 50, and change values for temperature in range (0, 1). We use the average of mAP@$K$ for y-axis, where $K \in \{1, 5, 10, 25, 50\}$.

We investigate the performance of different values of hyperparameters on CIRCO validation dataset (due to its managable size and diverse nature of images) using OpenCLIP as the retrieval model in Figure 2. From Figure 2 we inspect that retrieval performance is much impacted by temperature instead of top-$P$ or top-$K$. For higher temperature, the model will produce more diverse tokens, which hurts retrieval performance when matching with images, whereas the performance of top-$P$ and top-$K$ are very stable overall. The figure indicates that a lower temperature and a moderate top-$P$ with higher top-$K$ would result in better retrieval performance.

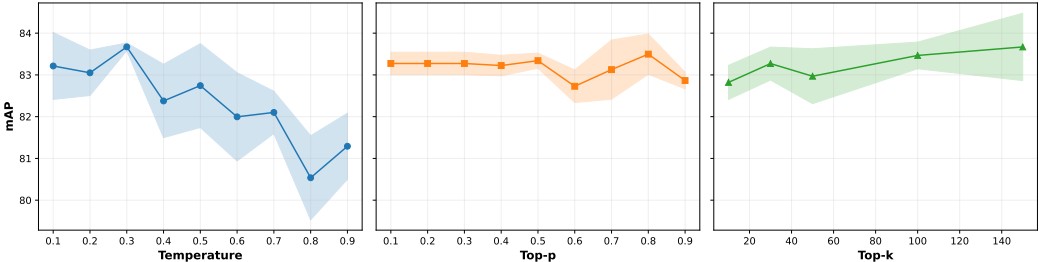

Figure 2: CIR performance (%) for 3 different runs via the MLLM under varying values for hyperparameters on CIRCO validation split. We use OpenCLIP with ViT-B/32 backbone to perform text-to-image task. Shadowed regions indicate standard deviation.

### 4.5.2 DIFFUSION MODELS

In the inference process of diffusion models, a strong correlation exists between key hyperparameters and the properties of the synthesized output. Specifically, a larger number of inference steps correlates strongly with the enhanced photorealism of the generated images. Furthermore, increasing the image guidance scale elevates the fidelity of the output to a given reference image. Conversely, a higher text guidance scale promotes stricter adherence to the input text prompt, often at the expense of output diversity. We use 7.5 for guidance scale, 3.0 for image guidance scale, and 30 for inference steps as base combination and only change one hyperparameter and fix the rest.

We show the retrieval performance when using different values for hyperparameters in diffusion models on CIRCO validation dataset with and without using MLLM in Figure 3. From Figure 3 we can see that using MLLM generated descriptions for generating images improves the retrieval performance for all three hyperparameters, which emphasizes the importance of using MLLM-generated descriptions as prompts instead of the original captions from datasets. We also show qualitative results of using MLLM for diffusion models in Appendix F. From the synthesized images, we observe more distinguishable artifacts when directly using raw modifications from CIRCO dataset.

The performance gap of using and not using MLLM is increasing for the guidance scale while the gap is decreasing with more inference steps, and the performance gap is almost consistent with varying values for image guidance scale. Across three hyperparameters, image guidance scale has

the most significant impact on retrieval performance, and higher values cause much worse performance. With larger image guidance scale, the generated images would be more like original input images instead of intended target images, thus deviating from target images and leading to worse performance. This indicates the importance of low values for image guidance scale to achieve good performance when reformulating CIR task to image-to-image task. We also noticed that retrieval performance increases first and then drops for varying number of inference steps, as more inference step produces more photorealistic images but also adds more artifacts. For time efficiency, a moderate number of inference steps should be sufficient, as indicated from the figure. Finally, we acknowledge that carefully selecting values for hyperparamters is labor-intensive, as different datasets and models might perform differently for the same setting of hyperparameters.

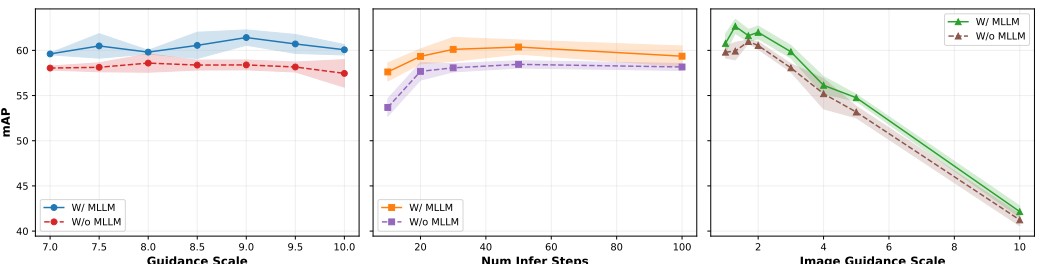

Figure 3: CIR performance (%) for 3 different runs via diffusion models under varying values of hyperparamters on CIRCO validation split *with* and *without* using composed descriptions from the MLLM as textual conditions. Raw captions from the dataset are used when not using MLLM. We use OpenCLIP with ViT-B/32 backbone to perform image-to-image task. Shadowed regions indicate standard deviation.

## 5 COMPUTE ANALYSIS

We analyze the computational overhead of the proposed framework for the CIR task. We focus on PEFUSE (T→I) and PEFUSE (I→I), as they involve both MLLMs and diffusion models. The former leverages MLLMs to compose target descriptions, while the latter employs diffusion models, either guided by MLLM-generated descriptions or by raw modifications, to generate target images. It is worth noting that both settings involve image-retrieval-based uni-directional conversion, without converting the generated target images into text. For consistency, we use the CIRCO validation split for both tasks across experiments. Specifically, we employ the lightweight Qwen2.5-VL-3B to examine the scaling effects of MLLMs, and the similarly sized LLaVA-1.5-7B-hf (Liu et al., 2024) to investigate the impact of different training strategies. Notably, Qwen2.5-VL-7B is trained with large-scale joint vision-language pretraining, whereas LLaVA-1.5-7B-hf aligns visual features with those of a frozen LLM. In addition, we include SDXL-Turbo (Sauer et al., 2024), a stable diffusion variant with a comparable parameter scale to SDXL-InstructPix2Pix, but supporting faster inference, thereby illustrating the trade-off between inference speed and output quality. We report the GPU memory consumption of each model during inference under `torch.bfloat16`, the average inference time per sample using the corresponding model, and the pipeline time, defined as the average end-to-end processing time per sample. The latter includes dataset loading, model loading, data generation, feature extraction, and performance evaluation. We adopt OpenCLIP for feature extraction in both PEFUSE (T→I) and PEFUSE (I→I), a batch size of 8, and the same MLLM sampling hyperparameters as in subsection 4.5. We apply identical settings for SDXL-InstructPix2Pix, while for SDXL-Turbo we use 60 denoising steps and an image guidance scale of 0.5, resulting in $60 \times 0.5 = 30$ effective inference steps.[1] Performance is evaluated using the average mAP across $\{1, 5, 10, 25, 50\}$.

We show the computational information in Table 5. In Table 5, for PEFUSE (T→I), we observe that Qwen2.5-VL-7B has the best performance among three MLLMs, where the same-sized LLaVA-1.5-7B-hf performs much worse than Qwen2.5-VL-7B, which stresses the importance of choosing

---

[1] https://huggingface.co/stabilityai/sdxl-turbo

proper MLLMs when converting queries to texts. Another important observations is that Qwen2.5-VL-7B surpasses Qwen2.5-VL-3B by 1.67% while being faster at inference time. On the other hand, we utilize Qwen2.5-VL-7B for PEFUSE (I→I), as it is optimal among 3 MLLMs. From Table 5, SDXL-Turbo is more than $3\times$ faster and outperforms SDXL-InstructPix2Pix by 2.27% when using raw modifications. When employing generated descriptions, similar trends are observed. This implies that using MLLMs can produce more performance gains for both models, whereas SDXL-InstructPix2Pix benefits more ($\uparrow$ 2.11%) than the SDXL-Turbo ($\uparrow$ 0.75%) from generated target descriptions.

Table 5: Compute analysis of the proposed pipeline on the CIRCO validation split over 3 runs. Inference time and pipeline time are reported per sample, with mean and standard deviation. SDXL-Instr denotes SDXL-InstructPix2Pix.

| Method | Models | Memory (MB) | Inference Time (s) | Pipeline Time (s) | Avg mAP (%) |
|---|---|---|---|---|---|
| PEFUSE (T→I) | Qwen2.5-VL-3B | 7161 | 0.77 ($\pm$ 0.06) | 1.03 ($\pm$ 0.09) | 81.52 ($\pm$ 0.23) |
| | Qwen2.5-VL-7B | 15816 | 0.44 ($\pm$ 0.04) | 0.69 ($\pm$ 0.07) | 83.19 ($\pm$ 0.36) |
| | LlaVA-1.5-7B-hf | 13472 | 0.73 ($\pm$ 0.05) | 2.10 ($\pm$ 0.16) | 73.41 ($\pm$ 0.47) |
| PEFUSE (I→I) | SDXL-Instr. | 6725 | 4.19 ($\pm$ 0.10) | 5.58 ($\pm$ 0.04) | 57.60 ($\pm$ 0.63) |
| | SDXL-Turbo | 6725 | 1.27 ($\pm$ 0.03) | 2.12 ($\pm$ 0.49) | 59.87 ($\pm$ 0.22) |
| | Qwen2.5-VL-7B + SDXL-Instr. | 15816 + 6725 | 4.74 ($\pm$ 0.16) | 6.11 ($\pm$ 0.05) | 59.71 ($\pm$ 0.11) |
| | Qwen2.5-VL-7B + SDXL-Turbo | 15816 + 6725 | 1.77 ($\pm$ 0.07) | 2.64 ($\pm$ 0.46) | 60.62 ($\pm$ 0.18) |

## 6 CONCLUSION

This work represents the first systematic exploration of pseudo-fusion for both uni-directional and bi-directional modality conversion within CIR. We empirically quantify the relationship between CIR performance and the critical hyperparameters of modern generative models, and conduct computational analysis when using MLLMs and diffusion models. Our results demonstrate that the challenge of CIR can be effectively reframed by converting heterogeneous modalities into a single, unified modality. This approach enables the use of standard single-query retrieval systems, either intra-modal or cross-modal, leveraging existing high-performance models in a plug-and-play manner without training new modules. Furthermore, our analysis establishes that reformulating the CIR task as text-to-image retrieval is a more effective strategy compared to other conversion modes, and emphasizing the importance of choosing proper MLLMs and diffusion models when converting queries. The strong performance of generative models in this pseudo-fusion role underscores their potential as a powerful tool for modality unification and points to a promising future for generative, model-based fusion methods in multimodal learning.

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

## A    LIMITATIONS AND FUTURE WORK

While our method demonstrates promising performance, and be highly compatible with techniques like LDRE (Yang et al., 2024) that further enhance the CIR performance, it is subject to several challenges inherent to its design, which at the same time present opportunities for future research. First, the integration of diffusion models and MLLMs for the pseudo-fusion of modalities can lead to the accumulation of biases and the propagation of errors through the pipeline. Consequently, chaining multiple models can be error-prone and presents a significant challenge for optimizing CIR performance. Second, our method considers non-preprocessed user-provided text modifiers as inputs to the MLLM. We posit that performance could be enhanced through advanced text paraphrasing,

structured formatting, or prompt engineering strategies. However, such techniques must be carefully designed to mitigate the inherent risk of LLM hallucinations. Third, diffusion models and MLLMs are highly sensitive to their respective hyperparameters, nevertheless, tuning these parameters is a labor-intensive process that may lack generalizability across diverse datasets.

Based on these limitations, we identify several promising directions for future work. Efforts could focus on developing robust integration techniques to minimize error propagation in multi-model pipelines and on leveraging more powerful yet lightweight foundation models. Furthermore, considering more complex scenarios remains a compelling long-term goal. For example, compositions involving multiple input images, longer text narratives, or additional modalities like video, can be investigated. Another promising avenue is to explore iterative, multi-round fusion of images and texts to generate progressively more refined and more accurate descriptions. Finally, we realize such conversion paradigms cause latency, therefore, to massively deploy interpretable CIR systems as such, well-performed, and light-weighted models should be developed.

## B  DATASETS

Table 6 shows the details of each dataset we used in our experiments. Due to broken links in the Fashion-IQ dataset, we are missing some images comparing to the original dataset. Due to this fact, we reproduce some results on the data we obtained. CIRR, CIRCO, and GeneCIS datasets are the same as the original ones.

Table 6: Public benchmarks used in our experiments. We use the validation split for FashionIQ and test splits for CIRCO, CIRR, and GeneCIS.

| Dataset | # of Queries | # of Candidates |
|---|---|---|
| Fashion-IQ (Shirt) | 1940 | 6181 |
| Fashion-IQ (Dress) | 1859 | 3648 |
| Fashion-IQ (Toptee) | 1867 | 5261 |
| CIRCO | 800 | 123403 |
| CIRR | 4148 | 2315 |
| GeneCIS (Focus Attr) | 2000 | 20000 |
| GeneCIS (Change Attr) | 2112 | 31680 |
| GeneCIS (Focus Obj) | 1960 | 29400 |
| GeneCIS (Change Obj) | 1960 | 29400 |

## C  RESULTS ON GENECIS

We show the experiment results on GeneCIS in Table 7. From the table, we find text-to-image still performs the best while other conversion modes still be competitive with the baselines.

## D  SCALING LAW ON CATEGORIES OF FASHION-IQ AND CIRR SUBSETS

We report results of scaling law for each category of Fashion-IQ dataset and the subsets of CIRR in Table 8 and that of GeneCIS in Table 9.

## E  TEXT RETRIEVAL RESULTS

Formerly, we reformulated the CIR to either image-to-image retrieval via diffusion model or text-to-image retrieval via MLLM. Now, we show the results when reformulating CIR to text retrieval tasks with additional conversion on targeting images using MLLM. We report results in Table 10, Table 11, and Table 12, respectively.

## F  QUALITATIVE RESULTS USING MLLM FOR DIFFUSION MODELS

We show the superiority of generated images based on MLLM-generated texts over raw captions for CIRCO validation split in Figure 4 and Figure 5. It can be observed that using raw captions to

Table 7: Performance (%) comparison on **GeneCIS** test split using different retrieval models via PEFUSE to convert reference images and modification texts to composed texts or to synthesized images. All the retrieval models use ViT-B/32 backbone. Best results in boldface while the second best underscored. †: results from original papers.

| Method | Retrieval Model | Focus Attribute | | | Change Attribute | | | Focus Object | | | Change Object | | | Average |
|---|---|---|---|---|---|---|---|---|---|---|---|---|---|---|
| | | R@1 | R@2 | R@3 | R@1 | R@2 | R@3 | R@1 | R@2 | R@3 | R@1 | R@2 | R@3 | R@1 |
| Pic2Word† | CLIP (ViT-L/14) | 15.65 | 28.16 | 38.65 | 13.87 | 24.67 | 33.05 | 8.42 | 18.01 | 25.77 | 6.68 | 15.05 | 24.03 | 11.16 |
| SEARLE† | CLIP (ViT-B/32) | 18.90 | 30.60 | 41.20 | 13.00 | 23.80 | 33.70 | 12.20 | 23.00 | 33.30 | 13.60 | 23.80 | 33.30 | 14.40 |
| CIReVL† | CLIP (ViT-B/32) | 17.90 | 29.40 | 40.40 | 14.80 | 25.80 | 35.80 | 14.60 | 24.30 | 33.30 | 16.10 | 27.80 | 37.60 | 15.90 |
| LinCIR† | CLIP (ViT-L/14) | 16.90 | 29.95 | 41.45 | 16.19 | 27.98 | 36.84 | 8.27 | 17.40 | 26.22 | 7.40 | 15.71 | 25.00 | 12.19 |
| LinCIR+CIG-XL turbo† | CLIP (ViT-L/14) | 16.80 | 29.70 | 40.90 | 15.91 | 28.88 | 37.45 | 8.37 | 17.35 | 25.10 | 7.86 | 15.46 | 24.29 | 12.24 |
| PEFUSE (T→I) | CLIP | 18.20 | 31.15 | 41.85 | 14.54 | 25.76 | 35.89 | 13.67 | 25.26 | 34.90 | 17.40 | 29.03 | 38.67 | 15.95 |
| | OpenCLIP | 19.55 | 31.95 | 43.70 | 14.73 | 26.28 | 36.51 | 17.96 | 28.27 | 38.06 | 16.73 | 30.97 | 41.33 | 17.22 |
| | SigLIP2 | 17.00 | 29.35 | 39.65 | 14.63 | 26.70 | 36.17 | 18.27 | 29.34 | 38.27 | 19.59 | 31.58 | 42.24 | 17.37 |
| PEFUSE (I→I) | CLIP | 15.90 | 27.15 | 37.50 | 11.03 | 20.36 | 27.60 | 9.18 | 18.88 | 27.35 | 8.16 | 17.60 | 27.65 | 11.07 |
| | OpenCLIP | 17.60 | 30.65 | 41.70 | 11.22 | 21.07 | 29.36 | 10.41 | 19.90 | 28.42 | 9.08 | 18.83 | 28.06 | 12.08 |
| | SigLIP2 | 17.95 | 29.15 | 40.00 | 11.17 | 21.69 | 30.21 | 10.20 | 19.49 | 27.91 | 9.90 | 20.71 | 30.71 | 12.31 |
| PEFUSE (T→T) | CLIP | 16.55 | 28.80 | 39.65 | 10.18 | 20.45 | 29.02 | 14.85 | 24.44 | 33.27 | 12.30 | 23.98 | 32.60 | 13.47 |
| | OpenCLIP | 16.60 | 30.90 | 41.95 | 12.26 | 22.44 | 32.81 | 16.53 | 27.35 | 35.87 | 16.17 | 27.91 | 37.96 | 15.39 |
| | SigLIP2 | 14.60 | 25.75 | 35.65 | 9.42 | 17.95 | 25.95 | 11.84 | 20.00 | 28.37 | 11.02 | 19.64 | 29.44 | 11.72 |
| PEFUSE (I→T) | CLIP | 17.45 | 30.70 | 41.00 | 10.94 | 21.07 | 30.45 | 10.97 | 19.74 | 28.16 | 11.58 | 20.82 | 29.64 | 12.74 |
| | OpenCLIP | 17.45 | 30.45 | 41.75 | 10.84 | 21.40 | 31.20 | 11.84 | 21.89 | 30.77 | 11.22 | 21.07 | 30.26 | 12.84 |
| | SigLIP2 | 17.40 | 29.60 | 39.80 | 9.90 | 20.69 | 31.11 | 11.22 | 20.61 | 28.78 | 10.51 | 19.80 | 29.54 | 12.26 |

Table 8: Scaling law on each category of **Fashion-IQ** and **CIRR** subsets using OpenCLIP for text-to-image task.

| Backbone | Shirt | | Dress | | Toptee | | CIRR | | |
|---|---|---|---|---|---|---|---|---|---|
| | R@10 | R@50 | R@10 | R@50 | R@10 | R@50 | $R_{subset}@1$ | $R_{subset}@2$ | $R_{subset}@3$ |
| ViT-L/14 | 29.54 | 47.01 | 25.23 | 43.73 | 33.69 | 54.37 | 73.49 | 88.82 | 95.08 |
| ViT-H/14 | 30.41 | 47.22 | 26.90 | 47.71 | 33.90 | 55.33 | 74.41 | 89.23 | 95.33 |
| ViT-g/14 | 30.52 | 48.87 | 25.12 | 46.05 | 34.82 | 55.44 | 74.15 | 89.57 | 95.45 |
| ViT-bigG/14 | 31.39 | 47.32 | 25.12 | 45.56 | 34.82 | 55.60 | 75.90 | 89.37 | 95.59 |

Table 9: Scaling law on each category of **GeneCIS** using OpenCLIP for text-to-image task.

| Backbone | Focus Attribute | | | Change Attribute | | | Focus Object | | | Change Object | | |
|---|---|---|---|---|---|---|---|---|---|---|---|---|
| | R@1 | R@2 | R@3 | R@1 | R@2 | R@3 | R@1 | R@2 | R@3 | R@1 | R@2 | R@3 |
| ViT-L/14 | 17.85 | 30.30 | 41.90 | 14.25 | 27.27 | 37.12 | 16.63 | 27.50 | 38.21 | 18.67 | 31.02 | 40.61 |
| ViT-H/14 | 19.10 | 31.35 | 43.50 | 15.58 | 27.18 | 37.97 | 17.50 | 28.16 | 37.09 | 17.81 | 29.59 | 39.54 |
| ViT-g/14 | 19.25 | 32.05 | 42.10 | 15.77 | 27.94 | 37.83 | 17.35 | 27.50 | 37.24 | 17.19 | 29.69 | 39.39 |
| ViT-bigG/14 | 19.00 | 31.15 | 42.95 | 16.57 | 28.79 | 39.44 | 17.55 | 27.91 | 37.45 | 18.83 | 30.77 | 40.87 |

Table 10: Performance (%) comparison on **Fashion-IQ** validation split using different retrieval models when additionally converting targeting images to texts. All retrieval models use ViT-B/32 backbone.

| Method | Retrieval Model | Shirt | | Dress | | Toptee | | Average | |
|---|---|---|---|---|---|---|---|---|---|
| | | R@10 | R@50 | R@10 | R@50 | R@10 | R@50 | R@10 | R@50 |
| PEFUSE (T→T) | CLIP | 14.33 | 25.57 | 8.82 | 20.55 | 16.01 | 29.57 | 13.06 | 25.23 |
| | OpenCLIP | 16.39 | 26.96 | 13.23 | 27.92 | 19.39 | 34.33 | 16.34 | 29.74 |
| | SigLIP2 | 2.53 | 5.88 | 1.72 | 5.38 | 3.05 | 7.61 | 2.43 | 6.29 |
| PEFUSE (I→T) | CLIP | 10.67 | 20.93 | 5.16 | 16.41 | 8.94 | 20.03 | 8.26 | 19.12 |
| | OpenCLIP | 14.28 | 26.55 | 8.18 | 19.96 | 12.32 | 25.44 | 11.59 | 23.98 |
| | SigLIP2 | 8.61 | 17.27 | 5.59 | 15.33 | 8.94 | 19.34 | 7.72 | 17.31 |

Table 11: Performance (%) comparison on **CIRR** test split using different retrieval models when additionally converting targeting images to texts. All retrieval models use ViT-B/32 backbone.

| Method | Retrieval Model | Recall | | | | | Recall_subset | | |
|---|---|---|---|---|---|---|---|---|---|
| | | @1 | @2 | @5 | @10 | @50 | @1 | @2 | @3 |
| PEFUSE (T→T) | CLIP | 21.81 | 31.13 | 45.06 | 56.72 | 78.51 | 59.47 | 79.66 | 90.48 |
| | OpenCLIP | 30.65 | 42.96 | 58.99 | 70.68 | 89.25 | 69.45 | 86.00 | 93.81 |
| | SigLIP2 | 16.07 | 24.19 | 37.40 | 47.81 | 68.82 | 52.77 | 72.39 | 84.65 |
| PEFUSE (I→T) | CLIP | 6.80 | 13.76 | 28.27 | 42.12 | 72.27 | 36.12 | 57.83 | 74.80 |
| | OpenCLIP | 7.40 | 15.86 | 31.71 | 45.16 | 74.29 | 36.10 | 58.72 | 75.47 |
| | SigLIP2 | 7.52 | 14.12 | 26.82 | 39.13 | 67.49 | 34.41 | 56.63 | 74.41 |

Table 12: Performance (%) comparison on **CIRCO** test split using different retrieval models when additionally converting targeting images to texts. All retrieval models use ViT-B/32 backbone.

| Method | Retrieval Model | mAP@5 | mAP@10 | mAP@25 | mAP@50 |
|---|---|---|---|---|---|
| PEFUSE (T→T) | CLIP | 7.71 | 7.97 | 8.76 | 9.26 |
| | OpenCLIP | 11.52 | 11.86 | 13.07 | 13.71 |
| | SigLIP2 | 6.68 | 6.56 | 7.14 | 7.48 |
| PEFUSE (I→T) | CLIP | 3.89 | 4.35 | 4.88 | 5.29 |
| | OpenCLIP | 4.10 | 4.62 | 5.26 | 5.74 |
| | SigLIP2 | 3.65 | 3.84 | 4.40 | 4.78 |

generate images incurs more distinguishable artifacts for both diffusion models, and SDXL-Turbo-generated images has more artifacts than those of SDXL-InstructPix2Pix.

## G  PROMPTS

We show the prompts used for MLLMs when generating composed descriptions based on reference images and text modifications. When using the prompts, images are converted to base64 format and then insert into the prompts. We stress that MLLMs can also be employed to generate multiple texts from different aspects or views per query like in LDRE Yang et al. (2024), which further enhances CIR performance at the cost of extra computational overheads.

---

**Fashion-IQ**

You are an expert at visual perception and imagination of fashion items. Given a reference image of fashion items and modification instructions, mentally apply the changes and produce an accurate and complete natural-language description of the resulting fashion items. The modifications may describe direct attributes (e.g., "solid white with buttons"), comparisons (e.g., "longer sleeves," "lighter in color"), combined attributes (e.g., "black with a red cherry pattern and deep V neckline"), or negations (e.g., "no lace design"). Image: `base64_image`. Here are the modification instructions: `caption`. Focus on the fashion item and its attributes such as type, color, pattern, material, shape, fit, and style details. Ignore people and background from the image. Avoid imaginary things. Be specific and objective so that I can find targeting images based on your description solely without knowing the reference image or modification instructions. Do not use vague comparative terms like 'same/different/smaller/larger/shorter/longer/unchanged', etc. Instead, you should specify these differences clearly, like: another color instead of red (if no specific targeting color is mentioned), and a clear sky (if mentioned) instead of unchanged sky, etc. Now, describe how the final fashion item looks after applying the modifications. Write in 1 to 3 coherent sentences.

---

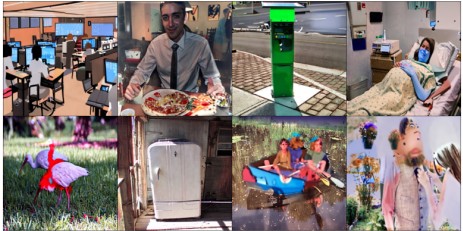
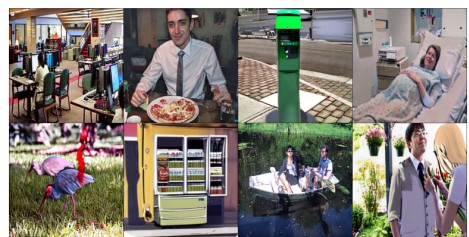

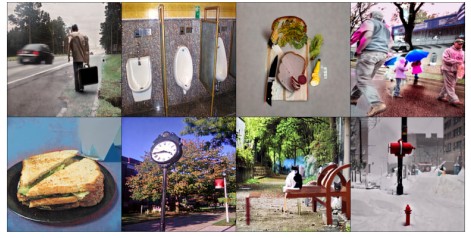
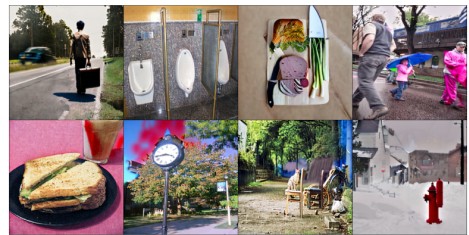

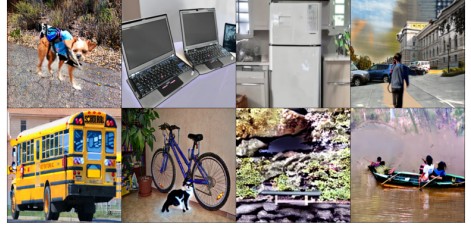
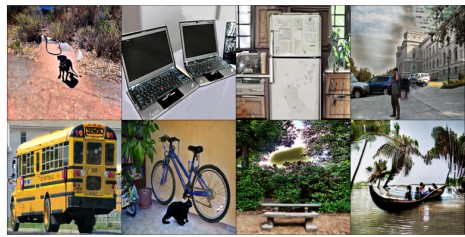

(a) With raw captions.  (b) With MLLM-generated descriptions.

Figure 4: Qualitative results of **SDXL-InstructPix2Pix**-generated images with Qwen2.5-VL-7B-generated descriptions and with raw captions from CIRCO validation split. We use 3.0 for image guidance scale, 7.5 for guidance scale, and 30 denoising steps.

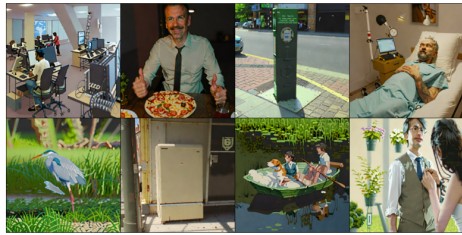
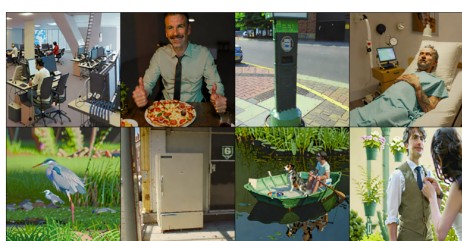

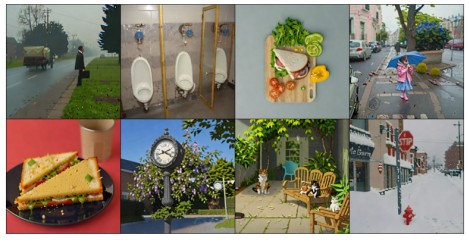
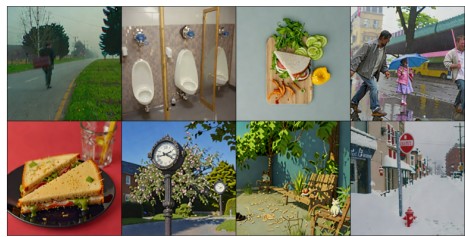

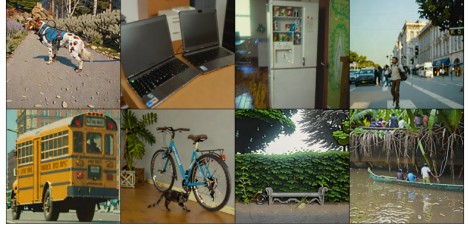
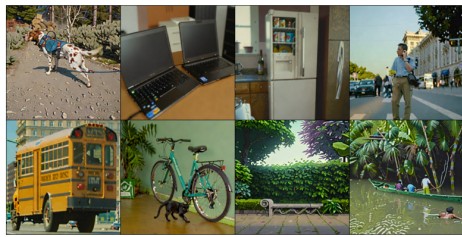

(a) With raw captions.  (b) With MLLM-generated descriptions.

Figure 5: Qualitative results of **SDXL-Turbo**-generated images with Qwen2.5-VL-7B-generated descriptions and with raw captions from CIRCO validation split. We use 0.5 for strength, 0.0 for guidance scale, and 30 denoising steps.

**CIRR**

You are an expert at visual imagination of real-world scenes. Given a reference image and modification instructions, mentally apply the modifications to the reference image and describe the resulting image in clear, complete English. Apply the modifications exactly as described, and ensure the final description reflects the scene after the changes. The modifications may include: 1. Cardinality: adjusting the number of objects (e.g., "only one bird remains"). 2. Addition: adding new objects or attributes (e.g., "add a red chair in the corner"). 3. Negation: removing elements (e.g., "remove the table"). 4. Direct Addressing: ensuring specific mentioned objects are clearly included. 5. Compare & Change: replacing one attribute with another (e.g., "same sofa but in leather"). 6. Comparative Statement: relative size, quantity, or intensity changes (e.g., "a larger group of people"). 7. Conjunction Statements: multiple modifications combined (e.g., "remove the tree and add two benches"). 8. Spatial Relations & Background: modifying positions, layout, or setting (e.g., "change the background to a beach"). 9. Viewpoint: adjusting perspective or framing (e.g., "zoom out to show the whole scene"). Image: `base64_image`. Here are the modification instructions: `caption`. Focus on the elements (like objects, people, and animals), their attributes (like color, size, shape, and quantity), spatial relations, and background. Avoid imaginary details and unnecessary repetitions. Be specific and objective so that I can find the targeting image from an image gallery based on your description solely without knowing the reference image or modification instructions. Do not use vague comparative terms like 'same/different/smaller/larger/shorter/longer/unchanged', etc. Instead, you should specify these differences clearly, like: another color instead of red (if no specific targeting color is mentioned), and a clear sky (if mentioned) instead of unchanged sky, etc. Write in 1 to 3 coherent sentences in English. Now, describe how the final image looks after applying the modifications.

**CIRCO**

You are an expert at visual imagination of real-world scenes. Given a reference image and modification instructions, mentally apply the modifications and produce an accurate, detailed description of the resulting scene. Apply the modifications exactly as described, and describe the final scene after the changes. The modifications may involve: 1. Cardinality: adjusting the number of objects (e.g., "has two boxes"). 2. Addition: introducing new objects or attributes (e.g., "a child under the umbrella"). 3. Negation: removing elements (e.g., "shows no bike"). 4. Direct Addressing: ensuring a specific object is present (e.g., "next to a window"). 5. Compare & Change: altering attributes (e.g., "different color," "surrounded by flowers"). 6. Comparative Statements: relative size, number, or intensity (e.g., "more stickers," "larger crowd"). 7. Conjunction Statements: multiple edits at once (e.g., "surrounded by snow and trees are more bare"). 8. Spatial Relations & Background: positioning or environment changes (e.g., "skyscrapers in the background"). 9. Viewpoint: changes in perspective or framing (e.g., "shot from above"). Image: `base64_image`. Here are the modification instructions: `caption`. Focus on the objects, people, animals, attributes (color, size, shape, quantity), spatial relations, and background context. Be specific and objective. Avoid imaginary details not supported by the reference image or the modification. Do not use vague comparative terms like 'same/different/smaller/larger/shorter/longer/unchanged', etc. Instead, you should specify these differences clearly, like: another color instead of red (if no specific targeting color is mentioned), and a clear sky (if mentioned) instead of unchanged sky, etc. Write 1 to 3 complete and coherent sentences so that I can find targeting images based on your description solely without knowing the reference image or modification instructions. Now, describe how the final image looks after applying these modifications.

**GeneCIS**

You are an expert at visual imagination of real-world scenes. Given a reference image and modification instructions, you should distinguish the objects and their attributes from the reference image, abd then mentally apply the modifications to the reference image, and describe the objects and their attributes in the final image after the changes. Image: `base64_image`. Here are the modification instructions: `caption`. Avoid imaginary details not supported by the reference image or the modification. Write 1 to 3 complete and coherent sentences so that I can find targeting images based on your description solely without knowing the reference image or modification instructions. Now, describe the objects and their attributes after applying the modification.

