# OpenReview forum: "Training-Free Pseudo-Fusion Strategies for Composed Image Retrieval via Diffusion and Multimodal Large Language Models"
_ICLR.cc/2026/Conference — Submitted to ICLR 2026_

### Official Review · Reviewer_bXvg · 2025-10-28

**Soundness:** 2
**Presentation:** 2
**Contribution:** 1
**Rating:** 2
**Confidence:** 4

**Summary:**

This paper proposes a PEFUSE, a training-free framework to address the CIR task. The proposed method utilizes diffusion models and multimodal large language models with uni-directional and bi-directional conversion to transform the CIR task to T2I and I2I tasks. Experiments validate the method achieves comparable performance compared with previous methods.

**Strengths:**

1.	The presentation of this paper is clear and easy for reading.

**Weaknesses:**

1. Insufficient contribution. The proposed method simply utilizes two existing tools, namely, MLLM and diffusion model, to transform the reference image and relative caption into a target caption or image. Using MLLM to generate target caption is a very old and existing technique, proposed in many previous literatures, such as CIReVL[1], LDRE[2]. Using the diffusion model based on the generated target caption, is very simple without any technical contributions, while the results also show the inferior performance, as stated in Table 1, 2, 3.

2. The proposed method show poor performance compared with previous works. For example, using CLIP as a fair comparison, the proposed PEFUSE performs poor compared with many old baselines, such as Pic2Word. If based on OpenCLIP or SigLIP2, please consider re-implement more previous SOTA methods for fair comparison, including literatures such as CIReVL [1], LDRE[2].

3. Time Consumption. As you mention that methods such as ImageScope, require more inference time with multiple models, it’ll be better to provide a detailed quantitative comparsion between your proposed method and others on inference time.

4. Ablation study is missing. More ablation study, such as the choice of MLLM, diffusion model, should be conducted.

5. Overall presentation. The presentation and experiments are conducted in a poor manner. Why only the results of uni-directional conversion appeared in the main text? More in-depth analysis should be conducted to show the relationship between uni-directional and bi-directional conversion. Also, the notation of ‘text-to-text’, ‘image-to-text’ between uni/bi-directional conversion should be defined separately, causing ambiguity for reading.

[1] Karthik, Shyamgopal, et al. "Vision-by-language for training-free compositional image retrieval." arXiv preprint arXiv:2310.09291 (2023).
[2] Yang, Zhenyu, et al. "Ldre: Llm-based divergent reasoning and ensemble for zero-shot composed image retrieval." Proceedings of the 47th International ACM SIGIR conference on research and development in information retrieval. 2024.

**Questions:**

Please refer to weaknesses.

---

> ### Author Response · Authors · 2025-12-03
>
> We express our gratitude to the reviewer for point out the weaknesses.
>
> ## Insufficient contributions
> Thanks for the comment. CIReVL and LDRE uses multiple models to generate multiple target descriptions based on reference images and modifications, which takes longer inference time, whereas our method only uses a single model to generate one caption and achieves better or comparable performance. As observed, our method converts CIR tasks to 4 single-modality retrieval tasks, which is never systematically done before. Our experiments empirically conclude that converting queries to texts is more effective than other conversion modes, and our framework is compatible with the techniques like in CIReVL and LDRE.
>
> ## Performance of our proposed framework
> We should note that CLIP does not always perform worse than Pic2Word. To further test the performance, we have run our method on GeneCIS as well. For Fashion-IQ and CIRR full sets, CLIP performs worse, while for CIRR subset, CIRCO, and GeneCIS it performs better. Especially for GeneCIS, CLIP performs much better than other baseline models. From our preliminary experiments, we found that model performance can collapse if not using a proper retrieval model. In our case, we found OpenCLIP is the optimal among 3 selected retrievers.
>
> ## Time consumption
> Thanks. To know more about the inference time and how each component impacts our framework, we measure performance and time on multiple MLLMs and diffusion models in table 5. We provide additional information about CIReVL as well: 4.13s for inference and 10.29s for the whole pipeline per sample on CIRCO validation split, which is significantly longer than ours. ImageScope uses LLaVA-v1.6-vicuna-7B as captioner, LLaMA3-8B-Instruct as reasoner, PaliGemma-3B-mix-224 as verifier, and  InternVL2-8B as evaluator, except VLMs like CLIP, and it takes much longer time for inference per sample than ours, as we only use Qwen2.5-VL-7B-Instruct for text-to-image.
>
> ## Ablation
> Please note that we have added extra information in section 5: Compute analysis. From the results, we confirm that the choice of both MLLMs and diffusion models impact retrieval performance, and conclude that MLLMs with better visual-lingual understanding (e.g., Qwen2.5-VL-7B-Instruct) improve performance while performance benefits more from faster diffusion models like sdxl-turbo.
>
> ## Presentation
> We only put results from uni-directional conversion in the main paper due to limited space, and  to show the extensiveness of our experiments, we additionally test our framework on GeneCIS dataset. For the relationship between uni-directional and bi-directional conversions, uni-directional conversion converts reference images plus text modifications to human-interpretable texts via MLLMs or images via diffusion models, and then match with the target images; based on the generated data of uni-directional conversion, bi-directional conversion generate descriptions of target images, and then use the generated texts or images to match generated texts. For clarification, we have added more explanation in methodology and refined the figure of the pipeline to eliminate potential ambiguity.

---

### Official Review · Reviewer_tXSS · 2025-10-31

**Soundness:** 2
**Presentation:** 2
**Contribution:** 2
**Rating:** 2
**Confidence:** 5

**Summary:**

The paper proposes a training-free *‘pseudo-fusion’* approach: by converting (reference image + modified text) into a single modality (text or image), the composite retrieval problem is transformed into a conventional unimodal retrieval task. This approach then uses a Diffusion model and a Multimodal LLM (MLLM) to achieve *‘unidirectional/bidirectional’* conversion. Experiments on **Fashion-IQ**, **CIRR**, and **CIRCO** demonstrate that text-to-image conversion generally outperforms other conversion methods and is competitive with some trained methods in several zero-shot scenarios. Overall, this is a systematic study of combining existing powerful generative and multimodal models to achieve zero-shot CIR.

**Strengths:**

1. **Comprehensive experimental design.** A systematic experimental design was implemented using multiple datasets (**Fashion-IQ**, **CIRR**, and **CIRCO**), multiple search engines (**CLIP**, **OpenCLIP**, and **SigLIP2**), and multiple transformation modes ($\(T \rightarrow I\$), $\(I \rightarrow I\)$). Scale and hyperparameter sensitivity analyses were also performed. This high coverage facilitates understanding of the impact of different factors.

2. **Insightful empirical conclusions.** Author found that text\(\rightarrow\)image conversion is generally good, with **OpenCLIP** outperforming **CLIP** in this scenario; Diffusion model artifacts lead to poor \(I \rightarrow I\) performance, etc. These are all useful guidelines for subsequent work.

**Weaknesses:**

1. **Relatively Incremental Academic Contribution.** As described in the paper, the method is primarily a combination of existing modules. To gain stronger acceptance, at least one or more clear new mechanisms should be emphasized and validated, or deeper theoretical/analytical support should be provided.

2. **Insufficient Consideration of Resources, Latency, Cost, and Privacy.** The model relies on large-scale MLLM and diffusion models (such as **Qwen2.5-VL** and **sdxl-instructpix2pix**), resulting in high runtime costs, inference latency, and memory usage (especially in large-scale retrieval, when multiple images/text must be generated for each query). The paper lacks an analysis of inference time, GPU/CPU costs, or deployability.

3. **Insufficient Ablation and Control Experiments.** There is a lack of comparison between different modules, as well as ablation analysis of caption generation or MLLM prompts.

4. **Insufficient Methodology and Analysis of Related Methods.** The two retrieval pipelines (MLLM-based and Diffusion Model-based) in the paper are not described in detail and appear to be an aggregation of existing methods.

**Questions:**

1. **Lack of qualitative visualization and failure case analysis.** The paper does not include qualitative visualizations of retrieval results or analyses of failure cases, which are important for illustrating the model’s strengths and limitations.

2. **Insufficient model efficiency analysis.** The use of large MLLMs and Diffusion models significantly increases retrieval overhead. It is recommended to compare the inference cost of the proposed method with related approaches to demonstrate the model’s feasibility and efficiency.

3. **Missing ablation studies.** It is unclear whether the prompt design affects the quality of the generated text/image and the final retrieval performance. Does image generation actually improve retrieval compared to directly using MLLM-generated text? If the $text\(\rightarrow\)image$ mode performs best, does that imply the image generation step is essential? Although experiments show that $text\(\rightarrow\)image$ conversion achieves the best performance, the underlying mechanism is not sufficiently explained.

4. **Insufficient methodological details and comparison.** The paper does not provide enough information on the specific implementation of retrieval based on MLLMs and Diffusion models. What is the difference between the Diffusion Model-based pipeline and the IP-CIR [2] or CIG [1] methods? Similarly, how does the MLLM-based pipeline differ from OSrCIR [3] or CoTMR [4]?

5. **Unclear justification for comparable performance.**  The $I \rightarrow T$ retrieval pipeline in the paper appears similar to OSrCIR [3]. OSrCIR [3] achieves a Recall@5 of 67.25 on the CIRR dataset using the ViT-G/14 backbone, while this paper reports comparable performance using the smaller ViT-B/32 backbone. Please explain the reason for this result.

[1]. L. Wang. Generative Zero-Shot Composed Image Retrieval.
[2]. You Li, Fan Ma, and Yi Yang. Imagine and Seek: Improving Composed Image Retrieval with an Imagined Proxy.
[3]. Yuanmin Tang, Xiaoting Qin, Jue Zhang, Jing Yu, Gaopeng Gou, Gang Xiong, Qingwei Ling, Saravan Rajmohan, Dongmei Zhang, and Qi Wu. Reason-before-Retrieve: One-Stage Reflective Chain-of-Thoughts for Training-Free Zero-Shot Composed Image Retrieval.
[4]. Zelong Sun, Dong Jing, and Zhiwu Lu. CoTMR: Chain-of-Thought Multi-Scale Reasoning for Training-Free Zero-Shot Composed Image Retrieval.

---

> ### Author Response · Authors · 2025-12-03
>
> Thanks for the reviewer for carefully inspecting.
>
> ## Academic contributions
> Thanks. In our work, we converted CIR to 4 different single-modality retrieval tasks, which was never systematically conducted by previous papers. Although focusing on one of the conversion modes is beneficial, our focus is empirically investigating which path of conversion works best for CIR task.
>
> ## Resources, latency, and cost
> Thanks for the comment. To further facilitate the understanding of compute latency, we have added more results that report inference time, GPU memory, and model performance. Popular training-free methods from previous papers like LDRE, ImageScope, CoLLM, and CIReVL use more than one LLM/MLLM and generate more captions, which is the common issue of current training-free methods and significantly enlarge inference time as well.
>
> ## Insufficient ablation and control experiments
> Thanks for this advice. For this, we use 2 additional MLLMs with one more diffusion model to conduct the ablation study and report the quantitative results in our paper. To minimize the impact factors, we used the same prompts even for different MLLMs. Through our experiments, we observed that the prompts affect the generated texts and thereby the CIR performance. We tried many different prompts and used the most performant prompts through preliminary experiments. We provided our prompts in the appendix for more information. As for designing even better prompts, we believe that our framework is highly compatible with prompt engineering techniques, which can be found from massive NLP papers and might steer our focus of the paper.
>
> ## Insufficient methodology and related methods
> For this aspect, we add more explanation in the methodology section to illustrate more about our proposed pipeline. Our method uses multiple models and concludes that converting CIR task to text-to-image is by far the most viable mode, which can be supported by other training-free methods that convert queries to (pseudo) texts and aggregating models.

---

### Official Review · Reviewer_C4aY · 2025-11-01

**Soundness:** 3
**Presentation:** 3
**Contribution:** 3
**Rating:** 4
**Confidence:** 3

**Summary:**

The authors propose a training-free "pseudo-fusion" framework for Composed Image Retrieval (CIR) that converts the query (reference image+modifying text) into a single modality using off-the-shelf MLLMs (to text) and diffusion models (to images), which reduces the probem to text-to-image or image-to-image retrieval task.

**Strengths:**

- The paper presents an original training-free perspective on Composed Image Retrieval (CIR), leveraging existing multimodal models to reduce CIR to I2T/I2I problem.

- The method is systematically evaluated across multiple standard CIR benchmarks (Fashion-IQ, CIRR, CIRCO) and different backbones (CLIP, OpenCLIP, SigLIP2), providing a thorough empirical study that supports the framework’s generality and practicality.

- Writing: the paper seem to be well-written and comfortable to read.

**Weaknesses:**

- Missing baselines: While the authors report results with three retrieval backbones (CLIP, OpenCLIP, and SigLIP2) combined with their method, the paper lacks proper training-free baseline comparisons. Specifically, the zero-shot CIR performance of these backbones - e.g., directly combining SigLIP2 image and text embeddings for retrieval - should be reported for context. Such baselines are crucial to fairly assess the real contribution of the proposed pseudo-fusion mechanism in the absence of additional training.

- Backbone influence and attribution of gains: the comparison against prior training-free methods raises concerns about whether improvements stem from the proposed approach or simply from stronger underlying models. The use of a specific MLLM backbone (e.g., Qwen2.5-VL-7B-Instruct) may disproportionately influence results. It remains unclear how much of the reported performance gain would persist if previous methods were also paired with this newer MLLM. An ablation controlling for the backbone choice would clarify the true contribution of the proposed pseudo-fusion strategy versus advancements in foundation model capabilities.

- Figure clarity and visual communication: Although the paper is generally well written, Figure 1—which serves as the main illustration of the proposed method, is confusing and visually overloaded. The figure contains too many intersecting arrows and mixed settings, making it difficult to follow the data flow and distinguish between the different stages of the pseudo-fusion process. For instance, the meaning of the colored arrows passing through the retrieval model and the duplicated sets of arrows in each branch are unclear. A clearer, modular reorganization, separating the different modes or stages, would significantly improve reader comprehension.

**Questions:**

See above

---

> ### Author Response · Authors · 2025-12-03
>
> We thank the reviewer for precious time, efforts, and valuable comments.
> ## Baselines
> We thank the comments from the reviewer. We are not very sure about what ‘reported for context’ means. Our proposed framework converts CIR to single modality retrieval problems via MLLMs and diffusion model, compatible with other refinement techniques for generating texts and images. We do compare multiple models with training and include a few training-free methods. We only generate one text or image per query, while other training-free methods generate multiple texts or use multiple LLMs/MLLMs, therefore it is hard to fairly compare. Here, our focus is to state the viability of converting composed queries to a single interpretable modality at high level, extensively benchmarking the model performance, and compare which conversion mode works best by far, which might inspire future research to improve the poorly-performed conversion modes.
>
> ## Backbones and gains
> Thanks for the comment. In the revised version of the paper, we used more MLLMs and diffusion models to conduct ablation study and report performance of different combinations of models in our paper. We confirm from our results that PEFUSE (T→I) performs better than other conversion modes, while the choice of generative models also has an impact on the performance.
>
> ## Figure clarity and visual communication
> Thanks again for the comment. We find it is mostly misleading because of the arrows, which we have improved by adding labels. At the same time, we have used different colors and boxes to emphasize the difference between uni-directional and bi-directional conversions, accompanied by a clearer caption.

---

### Official Review · Reviewer_CCuD · 2025-11-01

**Soundness:** 3
**Presentation:** 3
**Contribution:** 2
**Rating:** 2
**Confidence:** 3

**Summary:**

In this paper, the authors focus on zero-shot composed image retrieval. Specifically, they leverage diffusion models and MLLMs to bridge different modalities, enabling both intra-modal and cross-modal retrieval. Experimental results demonstrate the effectiveness of the proposed method.

**Strengths:**

1. The proposed training-free method converts multimodal queries into a single modality, which can be directly integrated into existing retrieval systems.
2. Both unidirectional and bidirectional modality conversion paradigms explore the potential of applying diffusion models and MLLMs to the zero-shot CIR task.

**Weaknesses:**

1. The designed method primarily focuses on implementing various experiments, but it lacks in-depth analysis of how to derive high-quality text or images that are well-suited for the CIR task.
2. The generation of images and text for the query side significantly drives retrieval latency.

**Questions:**

As listed above

---

> ### Author Response · Authors · 2025-12-03
>
> We sincerely thank the reviewer for the comments. We have carefully considered each point raised and revised the paper accordingly.
> ## Deriving high-quality texts or images
> Thanks for your comments. Our proposed framework combines various components, and solves the CIR task through reframing CIR to 4 different single-modality retrieval problems, which is a high-level concept that is compatible with other techniques to enhance the performance. While deriving high-quality texts and images is interesting, it is not the current focus and beyond the scope of this paper.
>
> ## Latency issues
> We acknowledge that the latency caused by converting queries using MLLMs and diffusion models is one of the limitations, which is a common problem for training-free methods using either LLMs or MLLMs. For clearer understanding, we further added the compute analysis together with ablation in our revised paper to provide more insightful information for researchers.

---

### Author Response · Authors · 2025-12-03

We would like to express our gratitude to the anonymous reviewers for their valuable comments. We believe that our paper has improved significantly as a result. In the following, we address the questions raised and explain how we have implemented the changes in the revised manuscript. We have used a green font color to highlight the changes, to facilitate re-review.

---

### Meta-Review · Area_Chair_EwST · 2025-12-29

**Summary:**

The paper presents a training-free approach with a practical idea, but it faces fundamental critiques on novelty, experimental rigor, and practicality. The core issues are:

**1. Lack of Novelty & Insufficient Evidence‌**: \
The primary concern across all reviews is that the method is seen as a ‌straightforward combination of existing tools‌ (MLLMs for captioning, diffusion models for image generation) without introducing a clear, novel technical mechanism. The contribution is perceived as incremental, and the performance gains may be attributable to using newer, stronger foundation models rather than the proposed fusion strategy itself.

**2. Major Experimental & Evaluation Gaps‌**: \
**Missing Crucial Baselines‌**: The paper lacks a ‌zero-shot CIR performance baseline‌ for the retrieval backbones (CLIP, OpenCLIP, SigLIP2). This is essential to fairly evaluate the contribution of the pseudo-fusion method.\
**Unfair & Confounded Comparisons‌**: Improvements over prior methods (e.g., Pic2Word, CIReVL, LDRE) are questionable. For a fair comparison, previous state-of-the-art methods need to be re-evaluated using the same strong backbones (OpenCLIP, SigLIP2).\
**Inadequate Ablation Studies‌**: There is a severe lack of ablation experiments to analyze the impact of choices like the specific MLLM/diffusion model used, prompt designs, and to isolate the contribution of the fusion strategy from backbone improvements.

**3. Practical Limitations Overlooked‌**: \
The paper ignores the ‌high computational cost, inference latency, and memory footprint‌ associated with using large MLLMs and diffusion models for large-scale retrieval. There is no quantitative analysis of inference time or resource usage, which is critical for assessing deployability.

Overall, the paper requires ‌major revisions‌ before it can be considered for acceptance. The authors must significantly strengthen the claim of novelty, conduct fair and controlled experiments (with proper baselines and ablations), and provide a cost-benefit analysis of the method's practicality.

**Reviewer Concerns:**

The rebutttal is too simple, and the main concerns of four reviewers are not addressed.

**Reviewer Scores:**

According to the current rebuttal, the four reviewers would not increase their scores even if participated fully in the discussion.

---

### Decision · Program_Chairs · 2026-01-26

Reject